# Hippocampal Deletion of CB1 Receptor Impairs Social Memory and Leads to Age-Related Changes in the Hippocampus of Adult Mice

**DOI:** 10.3390/ijms24010026

**Published:** 2022-12-20

**Authors:** Michela Palmisano, Alessandra Gargano, Bolanle Fatimat Olabiyi, Beat Lutz, Andras Bilkei-Gorzo

**Affiliations:** 1Institute of Molecular Psychiatry, Medical Faculty, University of Bonn, 53125 Bonn, Germany; 2Institute of Physiological Chemistry, University Medical Center of the Johannes Gutenberg University Mainz, 55131 Mainz, Germany; 3Leibniz Institute for Resilience Research (LIR), 55122 Mainz, Germany

**Keywords:** CB1 receptor, hippocampus, ageing, social memory, rAAV-mediated deletion

## Abstract

Endocannabinoid system activity declines with age in the hippocampus, along with the density of the cannabinoid receptor type-1 (CB1). This process might contribute to brain ageing, as previous studies showed that the constitutive deletion of the CB1 receptor in mice leads to early onset of memory deficits and histological signs of ageing in the hippocampus including enhanced pro-inflammatory glial activity and reduced neurogenesis. Here we asked whether the CB1 receptor exerts its activity locally, directly influencing hippocampal ageing or indirectly, accelerating systemic ageing. Thus, we deleted the CB1 receptor site-specifically in the hippocampus of 2-month-old CB1^flox/flox^ mice using stereotaxic injections of rAAV-Cre-Venus viruses and assessed their social recognition memory four months later. Mice with hippocampus-specific deletion of the CB1 receptor displayed a memory impairment, similarly as observed in constitutive knockouts at the same age. We next analysed neuroinflammatory changes in the hippocampus, neuronal density and cell proliferation. Site-specific mutant mice had enhanced glial cell activity, up-regulated levels of *TNFα* in the hippocampus and decreased cell proliferation, specifically in the subgranular zone of the dentate gyrus. Our data indicate that a local activity of the CB1 receptor in the hippocampus is required to maintain neurogenesis and to prevent neuroinflammation and cognitive decline.

## 1. Introduction

Healthy ageing is associated with a decline in cognitive abilities due to the cumulative effect of increasing neuroinflammatory glial activity, elevated levels of toxic aggregates of macromolecules, decreasing synapse densities and neuronal loss [1]. Importantly, there is an enormous variance in the speed of brain ageing between individuals and also in the onset and intensity of age-related changes between brain areas [2]. The reason for this variability is not fully understood, but it is hypothesized that the activity of the endocannabinoid system (ECS) plays a significant role in this process [3,4]. 

The ECS consists of the G protein-coupled cannabinoid receptor type-1 (CB1) and type 2 (CB2), their endogenous ligands and their synthesizing and degrading enzymes [5]. The endogenous cannabinoids (endocannabinoids) including anandamide (AEA) and 2-arachidonoylglycerol (2-AG), are produced on demand by lipid precursors [6] and can serve as retrograde transmitters in neuron-neuron communication, but they also mediate neuron-glia interaction [7] and are probably involved in intracellular signaling regulating mitochondrial activity [8]. 

In the brain, CB1 is the most abundant G-protein coupled receptor present mostly in neurons [9], whereas the expression of CB2 is low and localized mainly on glial cells [10]. The expression levels of CB1 receptors in the brain are strongly cell-type specific. In the hippocampus, the majority of cholecystokinin-positive, and a subset of calbindin-positive GABAergic interneurons, contain an extraordinarily high level of CB1 receptors [11], whereas the glutamatergic principal neurons have only a moderate expression [12]. CB1 receptor is present at very low levels in astrocytes [13] or resting microglia [14], but its expression is up-regulated when activated. Endocannabinoids, by activating microglial cannabinoid receptors (mostly CB2), reduce the inflammatory glial effect driving the M1 microglial phenotype to the anti-inflammatory M2 or the resting M0 states [15]. Similarly, synthetic cannabinoids exert an anti-inflammatory effect on isolated microglia [16]. Furthermore, homeostatic, surveying microglia release 2-AG and anandamide, and its production increases upon activation. Because 2-AG up-regulates microglial expression of CB1, while AEA increases the expression of CB2 receptors [17], it is feasible that this step is responsible for the elevated cannabinoid receptor expression in activated microglia. This process helps to terminate the inflammatory reaction promoting the anti-inflammatory, protective microglial phenotype [18]. Failure in the termination of inflammatory processes significantly contributes to the pathogenesis of neuroinflammation during ageing [19] or in neurodegenerative diseases [20]. Indeed, a persistent pro-inflammatory environment activates chronically microglia and astrocytes [21] and leads to axonal degeneration, synaptic impairments [22] and eventually to cognitive deficits. Considering the neuroprotective effect of cannabinoids, it was suggested that the increased CB1 receptor density in the brain of Alzheimer’s disease (AD) patients is a compensatory effect to the enhanced cytotoxicity and thus can be used as a therapeutic target [20]. In vitro study showing that 2-AG protected primary pyramidal neurons against amyloid beta-induced cytotoxicity supported this hypothesis [23]. Moreover, in vivo preclinical studies demonstrated that chronic treatment with the phytocannabinoid CB1 receptor agonist Δ9-Tetrahydrocannabinol (THC), together with cannabidiol slowed down the development of Alzheimer-like pathology [24] and reduced the symptoms [25] in the advanced phase of the disease in the APP/PS1 transgenic mouse model of AD. In the clinical trials, however, only alleviation of some symptoms was observed in THC-treated AD patients [26]. In Parkinson’s disease, there is no definitive change in CB1 or CB2 receptor levels [27]. In preclinical animal models of the disease, the CB2 agonists JW015 and AM1241 as well as the CB1 agonists WIN 55,212-2 and HU210 showed neuroprotective and anti-inflammatory effects [20]. In humans in clinical trials, similarly as in AD, only improvement in some symptoms was reported by administering the synthetic cannabinoid nabilone [28] or cannabidiol [29]. Lastly, in multiple sclerosis (MS) the combination of analgesic and motor effects with the anti-inflammatory and neuroprotective effects of cannabinoids is especially beneficial. Post-mortem histological analysis of MS patients revealed that cannabinoid receptors were present in several cell types around the injured axons [30]. Pharmacological treatment with cannabinoids effectively reduces symptoms of pain and spasticity in MS patients, thus significantly improving their quality of life without altering substantially the disease development [31]. 

Despite having an important homeostatic role, the ECS activity is not stable throughout life but changes during pathological conditions [32,33] and in physiological ageing [34,35]. The activity of ECS in the brain declines in advanced age because of a decreased density of CB1 receptors [36], their reduced coupling with G-proteins [37,38] and diminished levels of 2-AG due to a lowered expression of its main synthesizing enzyme diacylglycerol lipase alpha [39]. Because of the neuroprotective and anti-inflammatory activity of cannabinoid signaling [18], its decline probably contributes to the age-related increase in pro-inflammatory glial effect [19]. Although the presence and intensity of these changes vary between brain areas, all of them are present and prominent in the hippocampus [8,39]. Changes in ECS activity can substantially influence the ageing process because cannabinoid signaling is involved in the regulation of key homeostatic processes identified as hallmarks of ageing [40]: it mediates intercellular communication [7], influences neuronal stem cell activity [41], mitochondrial activity [8] and regulates autophagy, thus proteostasis [42]. Indeed, deletion of the CB1 receptor gene (*Cnr1*) led to several signs of early onset systemic ageing in mice, such as skin and testis atrophy [3] as well as osteoporosis [43]. Behavioral tests revealed that *Cnr1* null mutants mice (referred to as *Cnr1*^–/–^) showed learning deficits already at the age of 6 months in the partner recognition test and impaired spatial and operant learning at the age of 12 months [44,45]. These cognitive deficits were associated with pro-inflammatory glial activity, enhanced accumulation of the ageing marker lipofuscin [46], and reduced neurogenesis as well as neuronal loss, specifically in the hippocampus [13,47]. All these ageing-related changes in the hippocampus could be responsible for the learning deficits observed in the *Cnr1*^–/–^ mice. However, whether exacerbated hippocampal ageing is a result of a locally reduced CB1 receptor activity or an indirect consequence of a generally accelerated systemic ageing is not known. In the present study, we asked whether the hippocampus-specific deletion of the CB1 receptor elicits similar learning impairments and enhanced pro-inflammatory glial activity in the hippocampus as in mice with the constitutive deletion of the receptor. 

## 2. Results

### 2.1. Validation of Site-Specific Deletion of CB1 in the Hippocampus 

To assess the level of rAAV-mediated deletion of the CB1 receptor in the hippocampus, we measured the *Cnr1* mRNA expression levels by RT-PCR and the CB1 protein levels by immunohistochemistry 4 months after the viral injection, in 6-month-old CB1^flox/flox^ mice. We found a significant down-regulation of *Cnr1* in rAAV-Cre-Venus injected mice (*p* = 0.0288) (Figure 1A), as well as a decreased positive area stained by CB1 in cornu ammonis 1 (CA1) (*p* = 0.0010), dentate gyrus (DG) (*p* = 0.0110) and cornu ammonis 3 (CA3) (*p* = 0.0291) regions of the hippocampus of rAAV-Cre-Venus mice, compared to their age-matched controls, injected with the empty vector (Figure 1B,C). We also measured the area covered by the reporter protein Venus and we found that it was the same between the two groups (Figure 1C). 

### 2.2. Microglia and Astrocytes Activity in the Hippocampus

We next asked if the deletion of the CB1 receptor in the hippocampus promoted any pro-inflammatory glial activity in rAAV-Cre-Venus mice.

Therefore, we analysed the density of ionized calcium-binding adapter molecule 1 (Iba1)-positive microglia and the area-covered by glial fibrillary acidic protein (GFAP)-positive astrocyte in the hippocampus. We found that in rAAV-Cre-Venus mice the density of Iba1 positive cells was significantly enhanced compared to the age-matched controls in all the three regions of the hippocampus (CA1: *p* = 0.0249; CA3: *p* = 0.0130; DG: *p* = 0.0093) (Figure 2A,B). On the other hand, the area covered by GFAP showed a prominent change only in the CA1 hippocampal region of rAAV-Cre-Venus mice (*p* = 0.0495) than the rAAV-Venus group (Figure 2C,D).

### 2.3. Hippocampal mRNA Expression of Pro-Inflammatory Cytokines

Activation of microglia and astrocytes is accompanied by increased secretion of pro-inflammatory molecules such as cytokines, chemokines and other inflammatory mediators that lead to a progressive state of chronic inflammation in the brain [48,49]. To determine if the CB1 receptor exerts a local anti-inflammatory activity in the hippocampus, we measured the hippocampal mRNA expression levels of tumour necrosis factor-alpha (*TNFα*), interleukin 6 (*IL-6*) and interleukin 1 beta (*IL-1β*) using Real-Time PCR in 6-month-old mice. Mice lacking the CB1 receptor in the hippocampus showed a significant up-regulation of *TNFα* mRNA expression, compared to their age-matched controls (*p* = 0.0192) (Figure 3A). On the other hand, mRNA expression analysis from *IL-6* and *IL-1β* did not reveal any difference between the two groups (Figure 3B,C). 

### 2.4. Cell Proliferation in the Subgranular Zone of DG and in the Subventricular Zone of the Lateral Ventricle 

We then investigated whether the deletion of the CB1 receptor in the hippocampus affected the neurogenesis in the DG. Analysis of Ki-67, a marker for cell proliferation, revealed that rAAV-Cre-Venus mice had a considerably decreased number of proliferating cells in the subgranular zone (SGZ) of the DG (*p* = 0.0072) compared to rAAV-Venus injected mice (Figure 4A,C). To test if the reduction in the cell proliferation observed in Cre-injected mice was due to a site-specific loss of the CB1 receptor, we next counted the number of Ki-67 positive cells in the subventricular zone (SVZ) of the lateral ventricle. We did not find any difference between the two groups, thus proving the local activity of CB1 (Figure 4B,C).

### 2.5. Neuronal Densities in the Hippocampus

Next, we stained neurons with the neuronal nuclear protein NeuN to evaluate if the number of neurons differed between the two groups. Interestingly, neuronal densities of rAAV-Cre-Venus mice and their age-matched controls were similar in all the three hippocampal regions (Figure 5A,B).

### 2.6. Evaluation of the Social Memory in the Partner Recognition Test 

To assess the social memory in the 6-month-old mice, we used the partner recognition test as the behavioural paradigm. In the first session of the test, the social preference of mice was measured as the total time spent exploring the partner or an object. rAAV-Venus control mice and rAAV-Cre-Venus mice with a site-specific deletion of *Cnr1* performed similarly, displaying a significant preference for the mouse over the object on both experimental days (day 1: rAAV-Venus *p* = 0.0001; rAAV-Cre-Venus *p* < 0.0001. Day 2: rAAV-Venus *p* = 0.0033; rAAV-Cre-Venus *p* = 0.0001) (Figure 6A,B). These findings suggest that the hippocampal CB1 receptor does not play a significant role in the sociability of mice. In session two, when we assessed the social novelty preference, rAAV-Venus mice showed a higher preference for the novel partner after a delay of 1 h (*p* = 0.0426) compared to rAAV-Cre-Venus mice (Figure 6C), as they spent significantly more time interacting with the unfamiliar mouse (*p* = 0.0212) (Figure 6E). Most importantly, their locomotor activity, measured as speed and distance travelled, was indistinguishable between the two groups (Figure 6G,H). Thus, these data suggest that mice with reduced hippocampal CB1 receptor activity cannot discriminate between familiar and novel partners due to a memory deficit. We also noted that after an inter-trial interval of 2 h, there was no significant difference in the interaction time between the two trials in the control group, suggesting they also failed to recognize the previously encountered mouse (Figure 6D,F). 

## 3. Discussion

Previous studies have suggested that ECS activity significantly influences the ageing process: reduced CB1 receptor activity in mice with constitutive or GABAergic neuron-specific deletion of *Cnr1* led to an early onset of age-related deficits [13], whereas pharmacological elevation of CB1 receptor activity in aged mice restored several symptoms of brain ageing [50]. Understanding the mechanism by which CB1 receptor activity influences cognitive ageing is necessary to utilize the potential anti-ageing effect of CB1 receptor activation [51,52]. Although most of the histological changes were restricted in the hippocampus of the *Cnr1*^–/–^ mice, it is not known whether it is due to a higher sensitivity of this brain area to a generally accelerated systemic ageing or due to the local effect of decreased CB1 receptor activity. Our study now clearly shows that an approximately 50% reduction in CB1 receptor levels is enough to elicit similar learning deficit and pro-inflammatory glial activity as observed in the constitutive knockouts, thus suggesting the importance of local effects. Previously, we showed that young (2–3-month-old) null mutants mice do not display any histological change in the hippocampus and their learning ability is at least as good as in wild-type mice in the spatial or working-memory tests, in operant or skill-learning paradigms [44] whereas their social [53] and object recognition memory [54] is even better than their wild-type siblings. In the 6-month-old age group, however, an increasing number of Iba1-positive microglia and elevated GFAP-positive areas suggested an increased pro-inflammatory glial activity in the hippocampus, whereas the cortex and striatum remained unaltered from these changes [13]. Importantly, we found the first sign of significant learning deficit in *Cnr1*^–/–^ mice in this age group, namely in the social recognition test [53] which was the most sensitive model to age-related changes in our hands. In older mice, as the recognition ability of wild-type animals also started to wane, the difference between the genotypes decreased and in older age groups practically disappeared [44]. Additionally, *Cnr1*^–/–^ mice reported several signs of accelerated bodily ageing: they displayed atrophy in the subdermal fat and testis earlier than their wild-type siblings [53]. Several lines of evidence showed that age-related changes in peripheral organs trigger or accelerate brain ageing, thus cognitive decline. During the pathophysiology of many age-related peripheral conditions such as osteoarthritis or chronic kidney disease, there is an increase in cellular senescence, while clearance of these senescent populations mitigated tissue damage [55,56]. Intriguingly, recent meta-analysis studies of ageing populations with either osteoarthritis or chronic kidney disease revealed a link between these clinical conditions to dementia, to support that they independently contribute to it [57,58]. It was demonstrated that blood-born factors mediate the effects of peripheral ageing on the brain because young blood reversed cognitive deficits and reduced neurogenesis in aged animals and vice versa [59,60]. Our results make this scenario that accelerated systemic ageing is responsible for the cognitive decline and ageing symptoms in the hippocampus now unlikely. We hypothesize, instead, that a local effect of the reduced CB1 receptor activity is responsible for the observed phenotype in *Cnr1*^–/–^ because a partial hippocampal deletion of the CB1 receptor has induced a similar partner recognition deficit as found in the null mutants. The motor activity or social preference did not change in the hippocampus-specific mutants, showing the specificity of the effect of reduced CB1 signaling on the cognitive domain. Although previous studies revealed that the ECS has an important role in social interaction [61] and reward [62] by influencing the mesolimbic dopaminergic system [63], we now show that a 50% reduction in hippocampal CB1 receptor activity does not influence this function. 

A characteristic histological change in the hippocampus after site-specific deletion of the CB1 receptor was the enhanced pro-inflammatory glial activity. Neurons exert an inhibitory control on the microglia through soluble factors [64] as well as through directly interacting proteins [65]. Deletion of a limited neuronal population, like basal forebrain cholinergic neurons, leads to an enhanced microglial pro-inflammatory activity and to disturbance in ECS signaling [66]. The ECS plays an important role in microglial activity regulation: cannabinoid receptor antagonists increase pro-inflammatory activity [67] whereas agonists decrease it [68]. It has been hypothesized that the ECS modulates microglial activity mostly indirectly, because cannabinoid receptors are expressed at very low levels in resting microglia [69,70]. As observed in a recent study, although present to a low extent, the CB1 receptor might influence microglial activity, because microglia-specific deletion of CB1 receptors led to a reduced reactivity to LPS in males, but not in females [71], without altering the behaviour. The fact that microglia-specific deletion of CB1 receptors did not lead to any peculiar phenotype, suggests that the effect of microglial endocannabinoids on autoreceptors is rather limited. Microglial cells synthesize both 2-AG and anandamide in an activity-dependent manner: when activated, they significantly increase the synthesis and release of endocannabinoids [17]. The released microglial endocannabinoids bind and activate mostly neuronal CB1 receptors. Cannabinoid signaling plays a crucial role in the communication between microglia and GABAergic neurons, which express exceptionally high levels of CB1 receptors [72]. GABAergic but not glutamatergic neuron-specific deletion of the CB1 receptor induced a similar increase in pro-inflammatory glial activity associated with learning/memory deficits, as previously observed in constitutive knockouts [47] or hippocampus-specific knockouts, as found in the present study. We, therefore, hypothesized that 2-AG produced by microglia activates the CB1 receptors on GABAergic neurons and provides a continuous feedback control on glial activity. If this feedback signaling is reduced, the inhibitory control loosens, allowing a more pro-inflammatory microglial phenotype. 

Thus, a decreasing expression of CB1 receptors during ageing may significantly impair the regulatory effect of the cannabinoid system on microglial activity and contribute to the enhanced pro-inflammatory glial activity in the ageing brain. 

In pathophysiological conditions like traumatic brain injury, ischemia or other causes of hypoxia, there is a long-term rise of glutamate levels in the synaptic cleft leading then to extrasynaptic glutamate spillover [73]. High glutamate levels, probably as a compensatory mechanism, leads to enhanced production of inhibitory endocannabinoids [74,75]. Glutamate release induces microglial process extension toward neurons [76], whereas 2-AG induces chemotaxis in microglial cells acting on their CB2 receptors [74,77]. 

Several lines of evidence suggest that intrinsic and extrinsic factors significantly regulate adult neurogenesis and ECS activity has a strong influence on it: pharmacological elevation of cannabinoid receptor activity could elevate the proliferative capacity of progenitor cells and survival of neurons [78], whereas genetic deletion of CB1 receptor [79,80] or Dagla [81] reduces neurogenesis in a cell-type specific manner [82]. Our results showing reduced neurogenesis locally, in the subgranular zone but not in the more distant subventricular zone, which was free from the AAV-mediated deletion of CB1 receptors, demonstrate the importance of local CB1 receptor signaling in the maintenance of neuronal progenitor cell proliferation.

It is noteworthy that despite the pro-inflammatory glial activity and a lower ratio of neurogenesis the number of pyramidal or granular cells did not differ between the groups. We cannot exclude, however, that in advanced age all these age-associated changes lead to reduced neuronal numbers as observed in 12-month-old constitutive knockouts [47]. 

We hypothesize that reduced neurogenesis and increased pro-inflammatory glial activity in the hippocampus of the site-specific CB1 receptor knockouts are responsible for their learning deficits. We, therefore, assume that endocannabinoid signaling influences brain ageing locally, through the modulation of neuron-glia communication and neurogenesis. 

## 4. Materials and Methods

### 4.1. Animals

All the experiments were performed with young (2–3 months) and middle age (6 months) B6.cg Cnr1 tm1.2Ltz (CB1^flox/flox^) male mice. Mice were group-housed in standard laboratory cages, under reversed light/dark cycle. Water and food were provided ad libitum. Care of the animals and conduction of all experiments were approved by the Landesamt fuer Natur, Umwelt und Verbraucherschutz Nordrhein-Westfalen (LANUV NRW; 81-02.04.2019.A421).

### 4.2. Stereotaxic Viral Injections

The site-specific deletion of the floxed *Cnr1* gene was performed through stereotaxic viral injections in 2–3-month-old CB1^flox/flox^ mice. Recombinant adeno-associated viruses (rAAV1/2) expressing the Cre recombinase and the reporter gene Venus (named as rAAV-Cre-Venus) or the empty vector containing the reporter gene for the control group (referred as to rAAV-Venus) were provided by the Viral Platform of the University of Bonn and were injected bilaterally in the dorsal and ventral hippocampus (coordinates for the dorsal hippocampus: −1.8, ±1.0, −1.4; coordinates for the ventral hippocampus: −3.1, ±2.7, −3.6). 30 min before starting the surgery, mice were intraperitoneally given 0.1 mg/kg buprenorphine. They were then anaesthetized in a box with a mixture of oxygen 3% and isoflurane 2.5% and head-fixed in a computer-driven (Neurostar) stereotaxic (Stoelting) injection device. The anaesthesia was maintained at 1% isoflurane until the end of the surgery. Ear bars were soaked with a lidocaine gel to reduce any sign of discomfort and the eyes were protected from drying with an eye-protective gel. After the confirmation of the anaesthesia by tail pinch with forceps, the skin was shaved and disinfected with povidone-iodine and a 10 mm incision was made with a sterile scalpel. The bregma and lambda were made white with swabs dipped in hydrogen peroxide 30%. Bilateral hippocampal injections (1 μL per injection site, 0.2 μL/min for 5 min) was made with a 10 μL Hamilton syringe. The needle was left in position for an additional 5 min before being slowly withdrawn. At the end of the surgery, the wound was stitched, mice were returned to the cages to recover and any signs of discomfort were carefully monitored. The days post-surgery animals received intraperitoneal injections with 4 mg/kg carprofen.

### 4.3. Partner Recognition Test

We evaluated the social memory using the partner recognition test in 6-month-old mice (*n* = 13 per group). The test was carried out in the active (dark) phase of the animals, between 9.30 a.m. and 2 p.m., in a dimly lit, low-noise environment. Mice were transferred to test rooms to adapt at least 20 min before starting the test. The test took place in an open-field arena (44 cm × 44 cm). The floor of the arena was covered with sawdust soaked with mice odour, so the mice felt familiar with the environment and their behavior was not affected by changes in anxiety levels [83]. For 3 consecutive days, the animals were habituated to the open field arena for 5 min. The experiments started on the fourth day and consisted of two different sessions. In the first session (trial 1 or social preference task), animals were put into the familiar arena, and they were exposed to either an object (an empty grid cage) or a younger male mouse, from the same strain and sex as the experimental mouse but from another cage. Partner mouse was placed in a grid cage to avoid any inter-male aggression. The activity of mice was videotaped for 6 min, and the time spent investigating the partner or the object was calculated by using Noldus Ethovision System software (NoldusInformation Technology, Wageningen, The Netherlands). In the second session (trial 2 or novelty preference task) after a variable interval (day 4—1 h, day 5—2 h) test mice were placed again in the arena with the two grid cages, but in this session, both cages contained a mouse, the previously seen mouse (familiar) and a new one (unfamiliar). The activity was videotaped for a total time of 3 min. The animals were left undisturbed between two experiments for at least 24 h. A significant reduction in the time the animals spent with social interactions in the second presentation was considered an indication that the animals recognized the partner. Recognition index (%) was calculated as follows: (Tunf − Tf)/(Tunf + Tf) × 100, where Tunf represents the time spent with an unfamiliar mouse and Tf is the time spent with the familiar mouse.

### 4.4. Real-Time PCR

Mice were perfused with ice-cold phosphate buffer solution (PBS), and the hippocampus was isolated, fresh frozen on dry ice and stored at −80 °C until further processing. Hippocampi (*n* = 8 per group) were lysed in TRIzol (Life Technologies), and total RNA was extracted according to the manufacturer’s protocol. Samples were treated with RNase-free DNase I (1 U/1 μg total RNA) to remove DNA contamination. The quality of the RNA was assessed by measuring the ratio of the absorbance at 260 nm and 280 nm using a Nanodrop 2000 Spectrometer (ThermoScientific). Probes with a 260/280 ratio of less than 1.9 were rejected. cDNAs were synthesized using the SuperScriptFirst-Strand Synthesis System for RT-PCR kit (Invitrogen Corp., Carlsbad, CA, USA) with random hexamer primers. Total RNA (0.6 μg) was used as starting material for cDNA synthesis. Differences in mRNA expression were determined in triplicate by custom TaqMan^®^ Gene Expression Assays (Applied Biosystems, Darmstadt, Germany): Mm00432621_s1 for *Cnr1*, Mm00446190_m1 for *IL-6*, Mm00434228_m1 for *IL-1β*, Mm00443258_m1 for *TNFα* and Mm01545399_m1 for *Hprt*, used as a reference gene to standardize the amount of target cDNA. Typically, a reaction mixture consisted of 1× TaqMan^®^ Gene Expression Master Mix (Applied Biosystems, Darmstadt, Germany), 2 μL cDNA and 1× Custom TaqMan^®^ Gene Expression Assay. Samples were processed in a 7500 Real-Time PCR Detection System (Applied Biosystems, Darmstadt, Germany) with the following cycling parameters: 95 °C for 10 min (hot start), 40 cycles at 95 °C for 15 s (melting) and 60 °C for 1 min (annealing and extension). Analysis was performed using the 7500 Sequence Detection Software version 2.2.2 (Applied Biosystems, Darmstadt, Germany) and data were obtained as a function of threshold cycle (Ct). Relative quantitative gene expression was calculated with the 2^−ΔCt^ method.

### 4.5. Immunohistochemistry

Mice (*n* = 3 per group) were anaesthetized with a cocktail of xylazine/ketamine solution were transcardially perfused with ice-cold PBS, followed by a 4% paraformaldehyde (PFA) solution and used for CB1, Iba1 and GFAP staining. A second cohort of mice (*n* = 4–5 per group) was additionally injected, perfused with PBS and half brain used for Ki-67 and NeuN staining. For NeuN staining, mice from the two cohorts were pooled together. Brains from both cohorts were isolated and placed in 4% PFA for 2 h at +4 °C. Afterwards, they were cryoprotected in 20% sucrose overnight under shaking at +4 °C, then snap-frozen on dry ice-cooled isopentane, and stored at −80 °C until further processing. Coronal slices of the hippocampus were serially sectioned at 18 μm using a cryostat at −20 °C (Microm HM500, Med GmbH) and mounted onto glass slides. Glass slides were kept at −80 °C until further use. For staining, frozen slides were left to dry for 30 min at 38 °C on a hot plate. After drying, slices were framed with a PapPen, washed in PBS for 5 min at room temperature and permeabilized in PBS containing 0.5% Triton X-100 for 1 h at RT (for Iba1/GFAP co-staining as well as for NeuN staining), and 0.5% Triton X-100 and 3% BSA. For CB1 staining slides were permeabilized in PBS containing 0.5% Tween 20. Non-specific binding was avoided by incubating slides for 2 h in PBS containing 3% bovine serum albumin (BSA) and 0.05% Triton X-100 (for Iba1/GFAP co-staining and for NeuN) and 0.5% Tween-20 (for CB1). For Iba1 and GFAP co-staining slides were incubated 24 h at +4 °C with primary antibodies rabbit anti-Iba1 (Wako, 1:2000 diluted in 0.3% BSA/0.05% Triton X-100 in PBS) and chicken anti-GFAP (Abcam, 1:1000 diluted in 0.3% BSA/0.05% Triton X-100 in PBS). For NeuN staining slides were incubated 24 h at +4 °C with the primary antibody guinea pig anti-NeuN (Synaptic System 1:1000, diluted in 0.3% BSA in PBS) and with the primary antibody rabbit anti-Ki-67 (Abcam, 1:300 diluted in 0.5% BSA in PBS). For the CB1 staining, slides were incubated for 48 h at +4 °C with the primary antibody rabbit polyclonal anti-CB1 (Calbiochem, 1:500 diluted in 3%BSA/PBS). Afterwards, slides were washed three times for 10 min in PBS at room temperature, followed by incubation with the respective secondary antibody for 2 h at room temperature. Secondary antibodies used were: AF488 donkey anti-rabbit (Life Technologies, 1:1000 diluted in 0.3% BSA/0.05% Triton X-100), AF647 goat anti-chicken (Life Technologies, 1:2000 diluted in 0.3% BSA/PBS), AF647 goat anti-guinea pig (Life Technologies, 1:1000 diluted in 0.3%/PBS) and AF647 donkey anti-rabbit (Life Technologies, 1:1000, diluted in 0.3% or 0.5% BSA/PBS). Slides were then washed three times for 10 min in PBS followed by milli-Q water for 1 min, mounted with 4′,6-diamidino-2-phenylindole (DAPI, Southern Biotechnology Associates, Birmingham, AL, USA) and covered. Venus autofluorescence was detected between 515 and 528 nm. Fluorescence images were acquired with an LSM SP8 confocal microscope (Leica, Wetzler, Germany) with a 20× or 40× objective lenses. Fiji software (NIH, Bethesda, MD, USA) was used for images quantification. Iba1 positive cells were counted in all three regions of the hippocampus and divided by the area of interest. Results are presented as the number of positive cells/μm^2^. For CB1, GFAP and Venus, results are presented as % of area covered. The signal intensity above the threshold was the same for all the probes within the two experimental groups. Neuronal densities were estimated defining the NeuN positive area within the stratum pyramidale for the CA1 and CA3 hippocampal regions and within the GCL for the DG. The number of Ki-67 cells was counted in the SGZ of the DG and in the SVZ of the lateral ventricle. Results are expressed as the number of positive cells/μm^2^. At least 4–6 slices per animal were analysed.

### 4.6. Statistical Analysis

Statistical analysis and data visualization were performed using GraphPad Prism software (Ver. 9.0.0., GraphPad Software, San Diego, CA, USA). Student’s *t*-test was used to detect differences between the two groups. For the social preference test two-way ANOVA (genotype and target as main factors), followed by Bonferroni multiple comparison test was performed. For cytokines expression analysis, ROUT test was used to exclude significant outliers. In the partner recognition test, two control mice that remained mostly immobile in the corners were excluded from the study. One animal where the expression of the CB1 receptor was not reduced by 50% of the mean CB1 receptor expression in control animals was excluded from further analysis. Data are presented as means ± SEM.

## Figures and Tables

**Figure 1 ijms-24-00026-f001:**
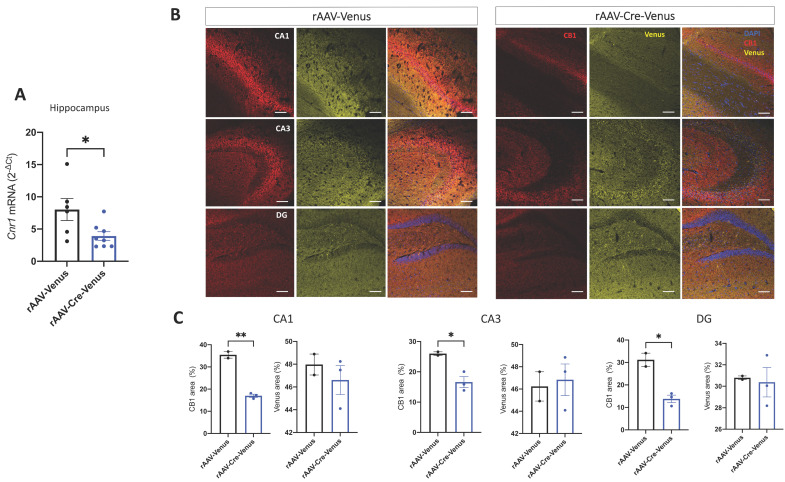
(**A**) mRNA expression levels of *Cnr1* in the hippocampus. (**B**) Confocal microscopy pictures representing the CB1 (red), Venus (yellow) and DAPI (blue) immunofluorescence and (**C**) quantification of the CB1-IR and Venus positive areas in the cornu ammonis 1 (CA1), cornu ammonis 3 (CA3) and dentate gyrus (DG) hippocampal regions of 6-month-old rAAV-Cre-Venus and rAAV-Venus (control) injected CB1^flox/flox^ mice. Scale bar 100 µm. Individual data points, mean value ± SEM are shown. Student’s unpaired *t*-test. * *p* < 0.05; ** *p* < 0.01.

**Figure 2 ijms-24-00026-f002:**
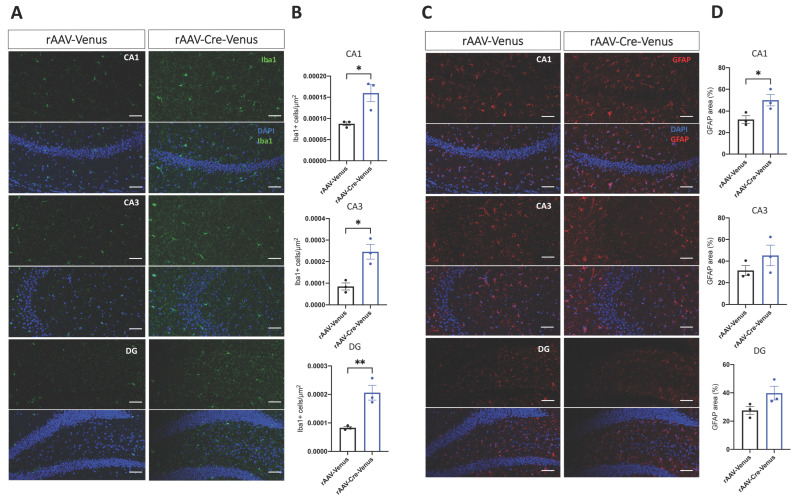
(**A**,**C**) Representative confocal photomicrographs of Iba1 (green), GFAP (red) and DAPI (blue) immunostaining in the cornu ammonis 1 (CA1), in the cornu ammonis 3 (CA3) and in the dentate gyrus (DG) regions of the hippocampus and (**B**,**D**) their quantification. Scale bar 50 µm. Individual data points and mean value ± SEM are shown. Student’s unpaired *t*-test. * *p* < 0.05; ** *p* < 0.01.

**Figure 3 ijms-24-00026-f003:**
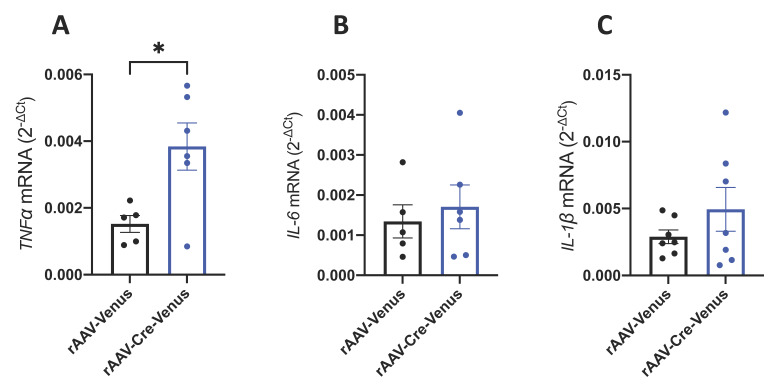
Hippocampal mRNA expression levels of the pro-inflammatory cytokines (**A**) *TNFα*; (**B**) *IL-6* and (**C**) *IL-1β*. Individual data points, and mean value ± SEM are shown. Student’s unpaired *t*-test. * *p* < 0.05.

**Figure 4 ijms-24-00026-f004:**
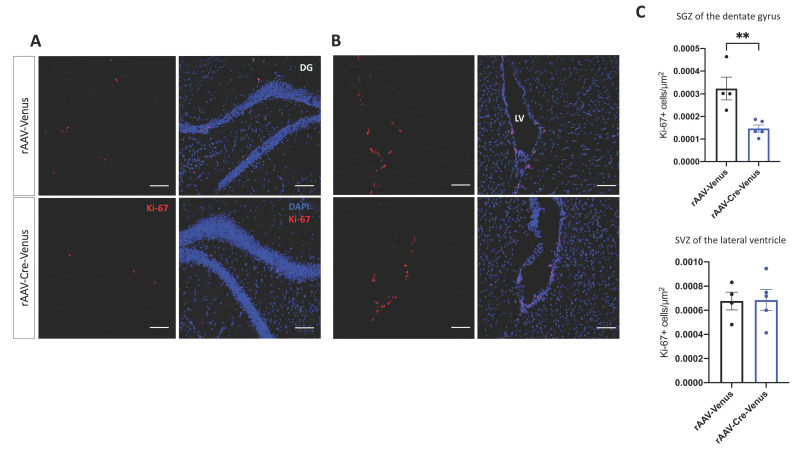
Representative confocal microscopy photomicrographs showing Ki-67 (red) and DAPI (blue) staining (**A**) in the subgranular zone (SGZ) of the dentate gyrus (DG) of the hippocampus and (**B**) in the subventricular zone (SVZ) of the lateral ventricle (LV) and (**C**) their quantification. Scale bar 50 µm. Individual data points and mean value ± SEM are shown. Student’s unpaired *t*-test. ** *p* < 0.01.

**Figure 5 ijms-24-00026-f005:**
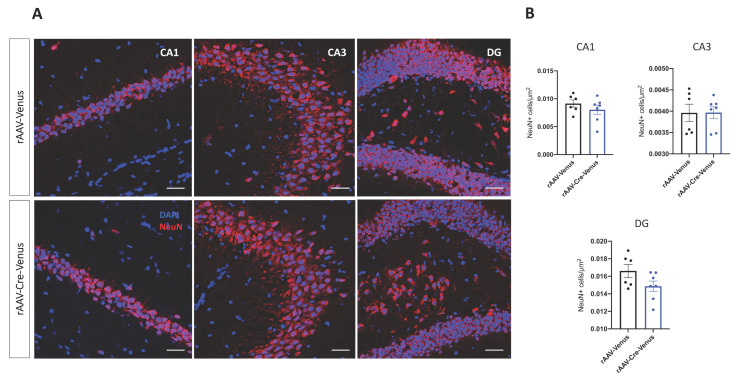
(**A**) Confocal microscopy pictures of NeuN (red) and DAPI (blue) immunofluorescence and (**B**) their quantification in the three regions of the hippocampus. Scale bar 25 µm. Individual data points and mean value ± SEM are shown. Student’s unpaired *t*-test.

**Figure 6 ijms-24-00026-f006:**
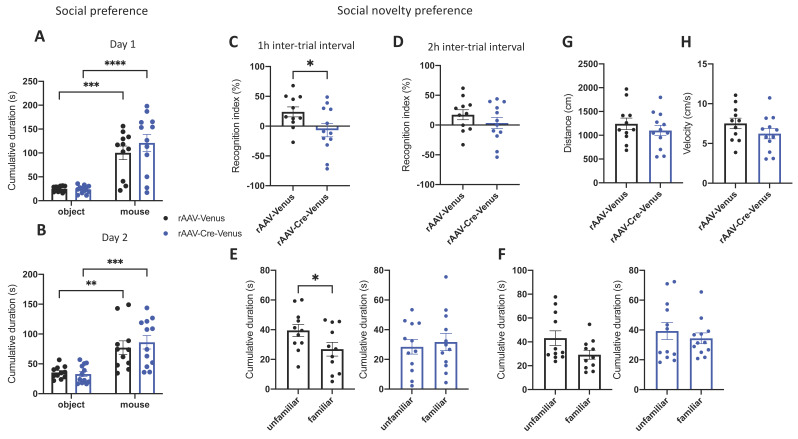
Social recognition memory in 6-month-old mice. (**A**) Social preference on Day 1 and (**B**) Day 2 of the social preference task. Social memory as social novelty preference (**C**,**E**) after 1 h on Day 1 and (**D**,**F**) 2 h on Day 2 inter-trial intervals. Locomotor activity of mice measured as (**G**) distance travelled and (**H**) velocity. Individual data points, and mean value ± SEM are shown. Two-way ANOVA, followed by Bonferroni post-hoc test for the social preference and Student’s unpaired and paired *t*-test for the social novelty preference. * *p* < 0.05; ** *p* < 0.01; *** *p* < 0.001; **** *p* < 0.0001.

## Data Availability

Data presented in this article are available on a reasonable request from the corresponding author.

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
