# Peer review of "Hippocampal Deletion of CB1 Receptor Impairs Social Memory and Leads to Age-Related Changes in the Hippocampus of Adult Mice"

_ijms, 2022, doi:10.3390/ijms24010026_

Round 1

Reviewer 1 Report

In this manuscript authors study the impact of hippocampus-specific deletion of the CB1 on learning impairments and enhanced pro-inflammatory glial activity in the hippocampus. The aim was to elucidate whether these changes are the same as in mice having ubiquitous knockout of the CB1.

Although the material is of high interest to the readers and clear presentation of the material several things raise concerns:

1) In the introduction there is the excessive self-citation of authors which is sometimes incongruous. It is necessary to provide the broad unbiased overview of the role of CB1/its agonists in neurodegenerative diseases, neuroinflammation and aging, primarily data for last 2-3 years.

2) Is well-known that the quantification of confocal microscopy images of  CB1 immunoreactivity (Fig. 1) in not reliable, unless the difference is high enough to be clearly visible by eye. Bars can be provided in case of detection with Western Blots or RT-PCR. Do authors have the data on Venus fluorescence in the same slices to demonstrate that it is the same in all analyzed samples?

2) The indicated should be expanded and revised carefully taking into the account recent data from other published studies: "2-AG produced by microglia activates the CB1 receptors on GABAergic neurons and provide a continuous feedback control on glial activity. If this feedback signaling is reduced, the inhibitory control loosens, allowing a more pro-inflammatory microglial phenotype [52]. Indeed, GABAergic but not glutamatergic neuron-specific deletion of the CB1 receptor induced a similar increase in pro-inflammatory glial activity associated with learning/memory deficits as found in constitutive knockouts [30] or hippocampus-specific knockouts as found in the present study."

3) Please discuss the postulated and provide the Ref for : "It has been hypothesized that the ECS indirectly modulates microglial activity because the expression of cannabinoid receptors (particularly CB1 receptors) is very low, if any, in microglia." This may add important information on neuroimmune interface in aging.

Minor/ for future:

In Stereotaxic viral injections it is indicated that the speed was 0.2 μl/min for 5 min. Taking into the account that AAV-Cre can induced some recombination outside the target area lower injection speed is recommended. 

Overall, the study is of high interest for readers and can be accepted after carefull consideration of abovementioned points

Author Response

We thank the reviewer for all the comments and suggestions that helped us to improve the manuscript. All the major changes are marked with grey. Please see the attachment.

Reviewer 2 Report

In this study the authors show the important role of cannabinoid receptor-1 (CB1) in social recognition memory of mice through hippocampal-specific deletion of CB1. Further analysis found significant changes of neuroinflammation, neuron density and cell proliferation. 

Overall, the authors show evidence that CB1 is required in the hippocampus to maintain neurogenesis and even prevent neuroinflammation and cognitive decline, however, the following concerns should be addressed by the authors:

MAJOR CONCERNS

  1. statistical stars are missing in most of the graphs. 

  2. AAV stereotypes should be mentioned in the method section. 

  3. General reporter staining of the AAV should be included in the figure to get an overview of the viral expression. 

  4. In the second result, the authors mentioned that the area with Iba1 was significantly enhanced. However, it is quite a general description. Is that because of the number of microglia, size of microglia, or the morphology of microglia? 

  5. Microglia activity should be further accessed by markers such as Cd68.  

  6. In the figure 4, SVZ is not a proper control, ideally you can distinguish the proliferation between transduced and non-transduced cells.

Author Response

We thank the reviewer for all the comments and suggestions that helped us to improve the manuscript. Please see the attachment

Round 2

Reviewer 1 Report

Authors significantly improved the Introduction and Discussion and added important information of other studies related to the finding. My comment on confocal images was also addressed and data on Venus fluorescence provided in the Results. I recommend to accept the manuscript after minor corrections

Minor:

1. Please provide the Refs for the papers that studied/postulated the following in the Discussion “Glutamate release induces microglial process extension toward neurons  […], ”

2. During proofing it is recommended to get rid of excessive use of the word “also” throughout the text

Author Response

We thank the reviewer for the positive comments and suggestions and especially the timely evaluation. We have now modified the manuscript  according to the last recommendations. Changes have been highlighted in red in the text.

Reviewer 2 Report

The authors have addressed my concerns. 

Author Response

We thank to the Reviewer the helpful critics with helped us the iprove significantly our manuscript and the timely review.